## Original Research Article

**Cite this article:** A.-K. Voss et al.
Environmentally dependent wood density
influences forest structure and dynamics in a
demographic vegetation model. *Quantitative
Plant Biology*, 7:e5, 1–13

DGVM; forest dynamics; LPJ-GUESS; wood
formation; wood density.

**Corresponding author:**
Anna-Kristina Voss;
Email: anna-kristina.voss@mgeo.lu.se

**Associate Editor:**
Dr. Félix Hartmann

# Environmentally dependent wood density influences forest structure and dynamics in a demographic vegetation model

Anna-Kristina Voss[1] , Stefan Olin[1], Hao Zhou[1,2], Patrick Fonti[3] and
Annemarie H. Eckes-Shephard[1]

[1]Department of Earth and Environmental Sciences, Lund University, Lund, Sweden; [2]Department of Earth and
Environmental Sciences, The Chinese University of Hong Kong, Hong Kong, China; [3]Swiss Federal Institute for Forest
Snow and Landscape Research WSL, Birmensdorf, Switzerland

## Abstract

Wood density (WD) is a crucial anatomical trait influencing forest carbon storage. However, dynamic global vegetation models (DGVMs) typically assume a fixed species-level WD, neglecting environment-driven variability. In this proof-of-concept study, we explore the potential impact of dynamic WD on tree- and forest-level carbon storage by integrating a simple temperature-response function of WD into the DGVM LPJ-GUESS from Smith et al., 2014.

Simulations along a temperature gradient show that incorporating environmentally responsive WD can substantially alter simulated stand structure and carbon stocks. Overall, our model experiments illustrated that sites with higher WD had more, but smaller trees, which stored less carbon compared to the standard model. The strongest effects were predicted to appear before canopy closure, where per-tree carbon deviated by up to 32%. This exploratory study suggests the need to represent a mechanism for dynamic WD to better assess ecological feedbacks to forest carbon storage predictions, particularly in young and regenerating forests.

## 1. Introduction

Forests act as crucial carbon sinks, mitigating climate change by absorbing and storing atmospheric carbon dioxide (Harris et al., 2021; Intergovernmental Panel on Climate Change, 2023). They play a fundamental role in the global carbon cycle, regulating atmospheric carbon concentrations and buffering against climate fluctuations (De Lombaerde et al., 2022; Pan et al., 2024). Within forests, carbon storage is determined by biomass accumulation and turnover (Friend et al., 2014; Körner, 2017; Pugh et al., 2020). A key factor in forest carbon storage is the investment of trees in woody biomass, which has compensated nearly 50% of humanity's recent fossil fuel emissions (Pan et al., 2024). Shifts in carbon allocation among leaves, stems and roots influence forest structure, composition and long-term carbon sequestration (Litton et al., 2007; Xia et al., 2017). These changes cascade through the ecosystem, affecting biodiversity, resilience to environmental stressors and overall forest functioning (Weiskopf et al., 2024).

At the individual tree level, carbon allocation is driven by species' growth strategy (Salguero-Gómez et al., 2016) and environmental constraints on physiological processes (Luo et al., 2015). Trees invest in different structures, affecting both their carbon storage capacity and overall functioning. However, carbon allocation is not static; it responds dynamically to environmental conditions. Factors such as temperature, precipitation and resource availability influence how trees distribute carbon between structural and non-structural components (Hartmann et al., 2020).

Wood density is a key trait in trees, linked to mechanical support, hydraulic efficiency and survival strategies, thereby shaping growth and lifespan (Ackerly, 2003; Chave et al., 2009; Cuny et al., 2019; Ogle et al., 2014). Wood density emerges through the process of wood formation (Rathgeber, 2017), which in temperate zones follows annual growth cycles, producing

distinct tree rings. Over the course of the growing season, trees first produce earlywood of lower density, composed of large-diameter, thin-walled cells. Later in the season, latewood is formed, characterized by smaller-diameter, thick-walled cells that contribute to higher wood density (Cuny et al., 2014). Environmental factors such as temperature and water availability influence both inter- and intra-annual variability in tree ring density (Begum et al., 2018; Buttò et al., 2020; Camarero & Hevia, 2020; Rosell et al., 2017). Additionally, as an adaptive trait, wood density varies spatially due to species differences, climate and ecosystem type (Mo et al., 2024; van der Maaten-Theunissen et al., 2013). It reflects a fundamental trade-off between growth and survival: trees with lower wood density grow faster but have shorter lifespans and are more vulnerable to damage, while those with higher wood density grow slower but are more resistant to mechanical stress and live longer (Chao et al., 2008; King et al., 2006). This trade-off influences forest dynamics by affecting mortality and resource competition, linking wood properties to forest demography (Chave et al., 2009).

Numerous empirical relationships between the environmental conditions and tree-ring level wood anatomical features, such as wood density, have been established and successfully applied in climate reconstruction. For example, positive correlations between latewood (LW) density and maximum density with late summer temperatures have been consistently reported, indicating the effect of temperature on LW formation (Björklund et al., 2017; Boakye et al., 2023; Cuny et al., 2019; Düthorn et al., 2016; Levanič et al., 2009; van der Maaten et al., 2012). While the transition from earlywood (EW) to LW has been studied, its controlling factors remain uncertain (Eckes-Shephard et al., 2022). Annual wood density features have been shown to shift with environmental change (Briffa et al., 1998; Pretzsch et al., 2018), with potential impacts for forest dynamics and carbon storage. Research has advanced significantly in understanding both the impacts (Micco et al., 2019) and environmental distribution (Mo et al., 2024; Yang et al., 2024) of wood density. However, despite this progress, major ecosystem models still do not fully integrate these mechanisms (Friend et al., 2022; Micco et al., 2019; Zuidema et al., 2018).

Dynamic global vegetation models (DGVMs) are widely used to simulate forest responses to climate change and predict global carbon storage. However, most DGVMs assume wood density is a fixed trait specific to plant functional types (PFTs). While this parameter helps determine growth strategies of PFTs, this simplification ignores climate-driven variations in wood density, potentially biasing simulations of tree size, biomass accumulation and forest dynamics. Consequently, DGVMs may not fully capture ecosystem carbon storage and forest responses to climate change, reducing the accuracy of global carbon cycle projections. Environmentally emergent wood density in DGVMs may therefore act as an important trait affecting growth at the level of the individual and the whole forest structure through forest–climate interactions, thereby influencing model predictions. Recent research suggests that static trait assumptions in DGVMs contribute to inaccuracies in forecasting forest responses to environmental change (Zuidema et al., 2018).

In this proof-of-concept study, we investigate the impact of environmentally emergent wood density on tree and ecosystem carbon storage. We incorporate a semi-empirical temperature-dependent wood density module into the state-of-the-art DGVM LPJ-GUESS (Smith et al., 2014), hereafter referred to LPJ-GUESS-WD. In this module, wood density emerges dynamically based on a non-linear response function between temperature and latewood density (Figure 1), influencing carbon allocation decisions within the model

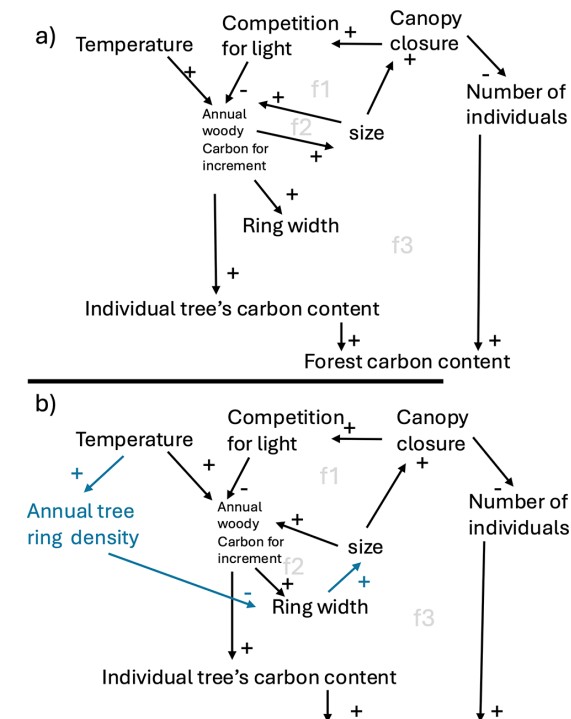

**Figure 1.** Processes that lead to individual tree and forest carbon content in the context of temperature-driven productivity and competition of light for (a) LPJ-GUESS-STD and (b) LPJ-GUESS-WD (S1.5.2). In a situation of forest recovery under standard conditions and no water limitations, LPJ-GUESS-STD has a direct relationship between temperature and an individual tree's carbon content, where the gain of carbon content per individual will only be slowed down once competition of light (i.e. potential carbon limitations) starts to influence the system (negative feedback f1). Ring width and size are independent of each other and emerge from a set of allometric rules, where these two variables do not directly determine each other. In LPJ-GUESS-WD (b), a new positive relationship between temperature and tree ring density constrains the width of the newly forming ring, with knock-on effects on other allocations such as height/size growth. Through the impact of varying annual tree ring density, ring width starts having a stronger impact on tree size (positive feedback (f2)). An amplifying positive feedback exists within this in the relationship between ANPP, ring width and size, which however is first dampened by temperature-dependent annual tree-ring density during early re-growth and later further dampened by competition upon canopy closure.

(Supplementary Material, 1.5.2). Mechanistic approaches to modelling wood formation processes are often complex and have been reviewed by Eckes-Shephard et al. (2022). For global-scale simulations in a DGVM, we chose to test this semi-empirical, parsimonious approach. In the standard version of LPJ-GUESS (hereafter LPJ-GUESS-STD), wood density is treated as a fixed parameter specific to each PFT. This means, for example, that the wood density of a simulated spruce tree in Siberia is the same as that of a spruce tree in Germany, despite their differing environmental conditions throughout their lifetime. In global simulations with limited PFTs of a coarse taxonomic resolution, dynamic wood density could further serve as a proxy for community-level changes in species composition, allowing the model to represent functional shifts that would otherwise be masked by fixed trait values. To address this limitation, LPJ-GUESS-WD simulates temperature-driven variation in wood density. This study is a proof-of-concept, aiming to illustrate potential implications of dynamic wood density for DGVMs and highlight directions for future model development. We hypothesize that this dynamic wood density affects tree growth and tree carbon content, with cascading effects on forest structure

and carbon storage, which vary depending on the tree life stage. To test this, we evaluate three key hypotheses:

- H1: Climate-driven wood density influences tree carbon content and stature. We test whether LPJ-GUESS-WD, which incorporates environmentally regulated carbon allocation to wood, alters carbon allocation patterns enough to impact tree stature.
- H2: Variable wood density at the level of an average individual per cohort affects tree number per plot. We hypothesize that changes in tree stature will influence ecological processes such as self-thinning competition, which may initiate at different forest ages when dynamic wood density is considered.
- H3: Climate change, through its influence on dynamic wood density, affects forest carbon dynamics differently depending on tree life stage. We test whether young and old forests respond differently to climate change (expressed as increase in temperature) when dynamic wood density is incorporated into model simulations.

## 2. Methods

### 2.1. LPJ-GUESS

The variable wood density module was integrated in the Lund-Potsdam-Jena General Ecosystem Simulator (LPJ-GUESS) (Smith et al., 2001; Smith et al., 2014), a dynamic global vegetation model (DGVM). LPJ-GUESS is designed to simulate tree-to-ecosystem processes, vegetation dynamics and carbon and nitrogen cycling under varying climatic and environmental conditions. LPJ-GUESS operates at a 0.5-degree gridcell resolution, receiving meteorological and soil property inputs. Vegetation is classified into PFTs, which are broad categories of plants sharing similar biome-specific, phenological and morphological features, and can also reflect distinct ecological growth strategies. The model has also been calibrated to the most common European tree species, and Norway spruce (*Picea abies*) is used in this study.

LPJ-GUESS simulates cohorts of trees, which are groups of trees established at the same time and sharing identical structural characteristics, such as diameter, height and crown area. One cohort consists of a varying number of trees and is modelled as one average tree, which we will refer to when talking about an individual tree. Tree biomass is allocated among four compartments: leaves, roots, sapwood and heartwood. Physiological processes such as photosynthesis, respiration, phenology, soil nutrient and carbon cycling and hydrology are modelled at daily time steps. In contrast, structural and demographic processes, such as biomass allocation, turnover, establishment and mortality occur annually. This version of the model has been enhanced with a new topography-related module (see Supplementary Section S1.5.1) which incorporates the effects of elevation on temperature.

In LPJ-GUESS-STD, wood density is treated as a fixed, species-specific parameter. Carbon allocation to stems, roots and leaves is optimized based on several allometric relationships and minimum allocation rules, all of which depend on wood density (see Supplementary Section S1.5.2). The model's climatic sensitivity, including change in stature, mainly results from the carbon input to the system (Figure 1). Consequently, warmer conditions may lead to increased carbon assimilation, which manifests as greater ring width and height. However, the carbon per unit volume remains constant across years, leaving an important dimension of plant

traits and allocation behaviour (e.g. the trade-off between wood density and tree height) unexplored.

Furthermore, the model does not allow for the exploration of climatic-driven changes in wood density and their influence on biomass allocation, tree height, and the cascading effects on the tree competition, and stand development.

By incorporating temperature-dependent wood density into the system, we expect that higher growing season temperatures will result in higher overall wood density. This creates a new trade-off that negatively impacts tree height growth as increased wood density limits carbon available for vertical growth (Figure 1, for a complete description of the implementation of LPJ-GUESS-WD, Supplementary Section S1.5.2.). Therefore, trees growing in warmer climates in LPJ-GUESS-WD will be relatively shorter but denser compared to those in LPJ-GUESS-STD. At the same age, carbon content per individual will then vary between sites due to (1) direct impact of wood density and (2) the indirect effect through differences in tree height.

### 2.2. The dynamic wood density module

We developed a temperature-dependent wood density module based on the well-documented relationship between temperature and LW density (Björklund et al., 2017; Boakye et al., 2023; Cuny et al., 2019; Düthorn et al., 2016; Levanič et al., 2009; van der Maaten et al., 2012) and initial data analysis (Supplementary Figure S2). Furthermore, we currently assume a constant EW to LW ratio of 70:30 within tree rings (Düthorn et al., 2016) and a fixed EW density of 139 kgC/m$^3$. While the latter two aspects are a simplification, it reflects the general pattern of a larger EW fraction in the ring (Pretzsch et al., 2014; Sandak et al., 2015), as well as the known mechanism of compensatory effects to achieve consistent cell anatomy during EW formation. We note that empirical evidence suggests EW:LW ratios may vary with temperature and site conditions, which will cause simulation bias on total ring density (TRD) response to temperature. Nevertheless, this simplified, semi-empirical approach allows us to explore, in a proof-of-concept manner, tree and forest response to wood density changes and highlight directions for future model development. Further, at the here explored sites, our approach is supported by the positive correlation of LW with overall tree ring density (Supplementary Figure S3). Importantly, the model structure is designed to be modular, allowing future replacement of the simplified earlywood component with more detailed, process-based formulations in future developments.

To construct the empirical response function, we used LW density data from 52 sites in the International Tree Ring Databank (National Oceanic and Atmospheric Administration, 2022) (Supplementary Table S1). We focused on spruce trees growing in Europe and Russia due to their well-established LW density–temperature relationship and data availability, but acknowledge that the data used might not be representative for spruce communities across Europe. To ensure consistency of the temperature-LWD signal, we tested that relationship across individuals, sites and species (S1.4). For model simulations, wood density values were converted into kgC/m$^3$ by multiplying by 0.828 to convert reported air-dry to oven-dry densities (Vieilledent et al., 2018) and 0.5 to account for the average carbon content in biomass (Thomas & Martin, 2012). Annual site-averaged values from 1901 to 1999 were used after outlier removal (z-score > 3; see Supplementary Section S1.2). Climate data were sourced from the CRU JRA data (University of East Anglia Climatic Research Unit

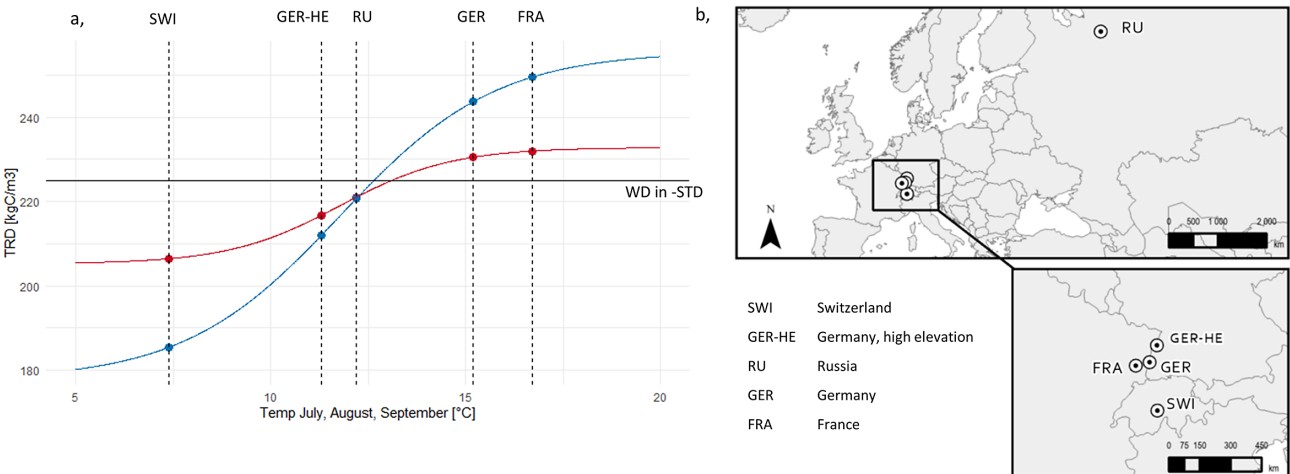

**Figure 2.** (a) Wood density functions modelling total ring density (TRD) resulting from the constructed latewood density functions (Red curve: fLWD-Best; blue: fLWD-Range), a constant earlywood density of 139 kgC/m³ and a fixed earlywood to latewood ratio of 70:30. Vertical dotted lines indicate the mean July+August+September temperature used in LPJ-GUESS for the five simulation sites. The horizontal line labelled WD in -STD corresponds to the fixed wood density parameter of 186 kgC/m³ in LPJ-GUESS-STD.
(b) Geographical locations of the simulation sites. A fully empirical temperature -TRD response function is fitted and discussed in Supplementary Section S1.3.1 and Supplementary Figure S6.

& Harris, 2020), with temperature values adjusted for elevation using a lapse rate of 0.649°C/100m, to match LPJ-GUESS (see Supplementary Section S1.5.1).

We identified July, August and September (JAS) as the period of highest correlation between LW density and temperature ($R^2$ = 0.34, p < 0.001; Supplementary Figure S2). To prevent overfitting to data-dense intervals, we binned data into 0.5°C temperature intervals and calculated mean LW density for each bin. A logistic function (Richard's curve) was fitted to the binned means using nonlinear least squares regression (nls()) in R (R Core Team, 2024), effectively capturing the physiological constraints in LW density response:

$$fLWD_T = a + \frac{b-a}{1 + e^{-c \bullet (T-d)}} \quad (1)$$

where $a$ and $b$ define the lower and upper asymptotes of LW density, $c$ is the growth rate, $d$ is the midpoint temperature, and $T$ is JAS temperature.

Since the best-fit function (fLWD-Best) does not capture the full observed range of LW densities (red line in Figure 2a), we also use an alternative function (fLWD-Range) (blue line in Figure 2a). This function maintains the same midpoint but adjusts $a$ and $b$ to the rounded minimum and maximum observed values, with $c$ re-fitted using nls(). fLWD-Range estimates the maximum plausible effect of variable wood density on carbon dynamics.

We further fit an empirical function for TRD dependent on the JAS temperature (Supplementary Figure S6). While more parsimonious, it is not suitable for exploring the ecological consequences of a wide range of TRD and therefore unsuitable for the scope of this research and the aim of a mechanistic model development (discussed in Supplementary Section S1.3.1).

### 2.3. Experimental runs

To assess the impact of dynamic wood density on tree- and forest-level carbon dynamics, we conducted stylised forest recovery simulations following a complete forest-destroying disturbance. We compared the LPJ-GUESS version with dynamic wood density LPJ-GUESS-WD to the standard version LPJ-GUESS-STD, which

assumes a fixed wood density of 186 kgC/m³. After a 700-year spin-up, the forests were completely destroyed, and left to recover for 300 years under constant climate. To isolate the effects of wood density on carbon stocks, all regenerating forests were replanted with 200 individuals per hectare of a single PFT: boreal needle-leaved evergreen trees, representing spruce. Only the cohort of replanted trees following the year of disturbance is considered in the result analysis. In LPJ-GUESS-WD, dynamic wood density was activated only after disturbance and set to the mean wood density based on the local climate (Supplementary Table S3). We ran two versions of the model, LPJ-GUESS-WD-Best and LPJ-GUESS-WD-Range (also WD-Best and WD-Range), which used fLWD-Best and fLWD-Range respectively to model wood density.

To analyse the effects of variable wood density on individual (H1) and forest-level (H2) carbon storage, both LPJ-GUESS-STD and LPJ-GUESS-WD were forced with a 'constant' climate derived from CRU JRA data (1901–1930) in a repeating 30-year cycle. To introduce variability in tree establishment, 30 climate ensemble members were generated, each starting at a different year within the cycle. Simulations were run at five sites (Figure 2), with 30 climate ensemble members at each site. Reported variables (i.e. height, Cwood/tree) are the mean of 25 replicate patches.

To test life-stage sensitivity to climate-driven wood density shifts (H3), we simulated a 2°C temperature increase starting at different forest recovery stages. The timing of warming was site-specific and determined by the canopy growth stage. This approach allows us to assess how wood density-driven changes affect trees at different age, size and canopy stages (Supplementary Section S1.6). A single climate ensemble member per site was selected to minimize variability, based on the lowest root mean square error in the canopy area time-series compared to the ensemble mean.

## 3. Results

Temperature-driven variations in wood density, both in sapwood and overall, diverged from the default values used in LPJ-GUESS-STD (Supplementary Figures S10 and S11). At GER and FRA, mean stem wood density (SWD) was simulated to be higher than the

default at 186 kgC/m$^3$, exceeding it by 5–7 kgC/m$^3$ in WD-Best simulations and by 8–12 kgC/m$^3$ in WD-Range simulations. In contrast, at all other sites, wood density was lower, with reductions ranging from 2 to 15 kgC/m$^3$ in WD-Best simulations and from 3 to 27 kgC/m$^3$ in WD-Range simulations.

These deviations align with site-specific temperature responses (Figure 1). FRA, which has the highest JAS temperatures, exhibited higher wood density, while SWI, with the lowest JAS temperatures, consistently showed lower wood density than the default. RU, located at the inflection point of the response function, showed no significant differences between the two response functions, although greater variability is observed in WD-Range. Higher variability was observed in sapwood density (Supplementary Figure S9), driven by the temperature response of the three most recent years and the corresponding stem width increment. However, SWD stabilized over time, as it reflects the mean temperature response across all growth years (Supplementary Figure S10). At all sites, except SWI, the model reproduced the described negative correlation between wood density and ring width (Supplementary Figure S12).

After 100 years of post-disturbance regrowth, the mean tree height in LPJ-GUESS-STD among the sites ranged from 16.79 m to 26.91 m, the mean carbon content in wood per tree (Cwood/tree) from 54.97 to 359.38 kgC, and the mean number of trees per hectare (trees/ha) from 45.33 to 140.03 (Supplementary Table S4). LPJ-GUESS-*WD*-Best deviated from LPJ-GUESS-STD in all three variables (Supplementary Table S4), but on an area-basis carbon mass per area was little affected, with LPJ-GUESS-*WD*-Best having 0.1 kgC/m$^2$ smaller difference across the gradient than LPJ-GUESS-STD. We subsequently report relative differences between LPJ-GUESS-STD and LPJ-GUESS-WD, to ease comparability between sites and to better highlight trends over time.

### 3.1. H1: Wood density influence on individual trees

Our proof-of-concept results show that the influence of dynamic wood density on Cwood/tree differs between LPJ-GUESS-WD and LPJ-GUESS-STD, showing both spatial and temporal variations across the study sites (Figure 3a). The largest variations were predicted to occur before canopy closure. At sites where wood density was lower than in LPJ-GUESS-STD, Cwood/tree was higher, while at RU, where wood density was close to STD, the change was minimal. In contrast, at sites with higher wood density than in LPJ-GUESS-STD, Cwood/tree tended to be lower, although this effect was less pronounced in GER and FRA when using WD-Best.

At all sites, Cwood/tree followed two distinct phases. In the early post-disturbance phase, LPJ-GUESS-WD simulations showed the greatest variability among ensemble members, with the largest deviations from LPJ-GUESS-STD. After canopy closure, the trajectories of the ensemble members stabilized around the mean, indicating a more uniform forest structure.

The mean deviations in Cwood/tree relative to LPJ-GUESS-STD reached over 41% for SWI, 7% for GER-HE, 2% for RU, −13% for GER and −17% for FRA in the WD-Range simulations. In contrast, using the WD-Best response function, mean deviations were smaller, reaching up to 16% for SWI, 4% for GER-HE, 4% for RU, −4% for GER and −6% for FRA (Figure 3a). Maximum deviations of individual ensemble members were even higher for both WD-Best and WD-Range at all sites.

Differences in mean tree height between LPJ-GUESS-WD and LPJ-GUESS-STD (Figure 3b) followed similar trends across both response functions, with more pronounced differences in the

WD-Range simulations. Sites with lower wood density than LPJ-GUESS-STD tended to develop taller trees, whereas sites with higher wood density than LPJ-GUESS-STD tended to develop shorter trees, particularly in the full-range response simulations (Figure 3b). Overall, deviations in mean tree height between LPJ-GUESS-STD and LPJ-GUESS-WD exhibited consistent trends both before and after canopy closure (Figure 3). For example, in LPJ-GUESS-WD, mean tree height in SWI and GER-HE exceeded LPJ-GUESS-STD by more than 14% and 3%, respectively (Figure 4a,b). Conversely, in GER and FRA, mean tree height was 5% and 7% lower, with individual ensemble members deviating by as much as −12% before canopy closure (Figure 4b,e). Meanwhile, RU showed only modest height differences between LPJ-GUESS-WD and LPJ-GUESS-STD.

### 3.2. H2: Wood density influence on forest dynamics

Following disturbance, the same number of trees was planted for all simulations, and tree numbers remain identical in LPJ-GUESS-WD and LPJ-GUESS-STD until canopy closure (Figure 4b). However, post-canopy closure, differences emerged, with the direction of these responses aligning with wood density deviations along the wood density–temperature gradient. Specifically, at sites where wood density was lower than in LPJ-GUESS-STD, self-thinning lead to greater tree loss, with up to 18 or 40 trees/ha in SWI and 13 or 16 fewer trees/ha in GER-HE in some climate ensembles of WD-Best or WD-Range, respectively. In contrast, at sites with higher-than-STD wood density, the number of trees increased in LPJ-GUESS-WD, with up to 10 more trees/ha for WD-Best and 23 trees/ha for WD-Range.

In GER and FRA, LPJ-GUESS-WD simulated shorter trees with denser wood (Supplementary Figure S10), leading to lower carbon storage per individual (Figure 3) but slightly more trees post-canopy closure (Figure 4b). Conversely, in SWI and GER-HE, the lower WD (Supplementary Figure S10) resulted in taller trees and therefore greater carbon storage per individual (Figure 3).

Before canopy closure, SWI, GER-HE, and RU stored more carbon per square meter (Cwood/m$^2$) (Figure 4a) than LPJ-GUESS-STD, while GER and FRA show the opposite trend, although this effect varies among ensemble members. Deviations from LPJ-GUESS-STD ranged from −20% to +26% with WD-Best and from −32% to +30% with WD-Range.

After canopy closure, this pattern reversed: SWI and GER-HE stored less Cwood/m$^2$, primarily due to fewer remaining trees, while GER and FRA stored more Cwood/m$^2$, especially in the WD-Range simulations, as they retained more trees with denser wood than LPJ-GUESS-STD (Figure 4). However, at most sites, over the course of 300 years the Cwood/m$^2$ eventually stabilized at little to no change compared to LPJ-GUESS-STD (Figure 4).

### 3.3. H3: Effects of climate change-driven wood density changes

A 2°C temperature increase and the resulting changes in wood density affected forests differently depending on their age and stage of canopy closure. Young forests, exposed to higher temperatures immediately after disturbance or before mid-canopy closure, exhibited an immediate and pronounced increase in mean wood density (light blue lines, Supplementary Figure S15). In contrast, older forests nearing or already at canopy closure, which have a history of lower SWD, responded more gradually as the effect of increased temperature on SWD unfolds over time. With a 2°C temperature increase, the emergent mean wood density shifted to

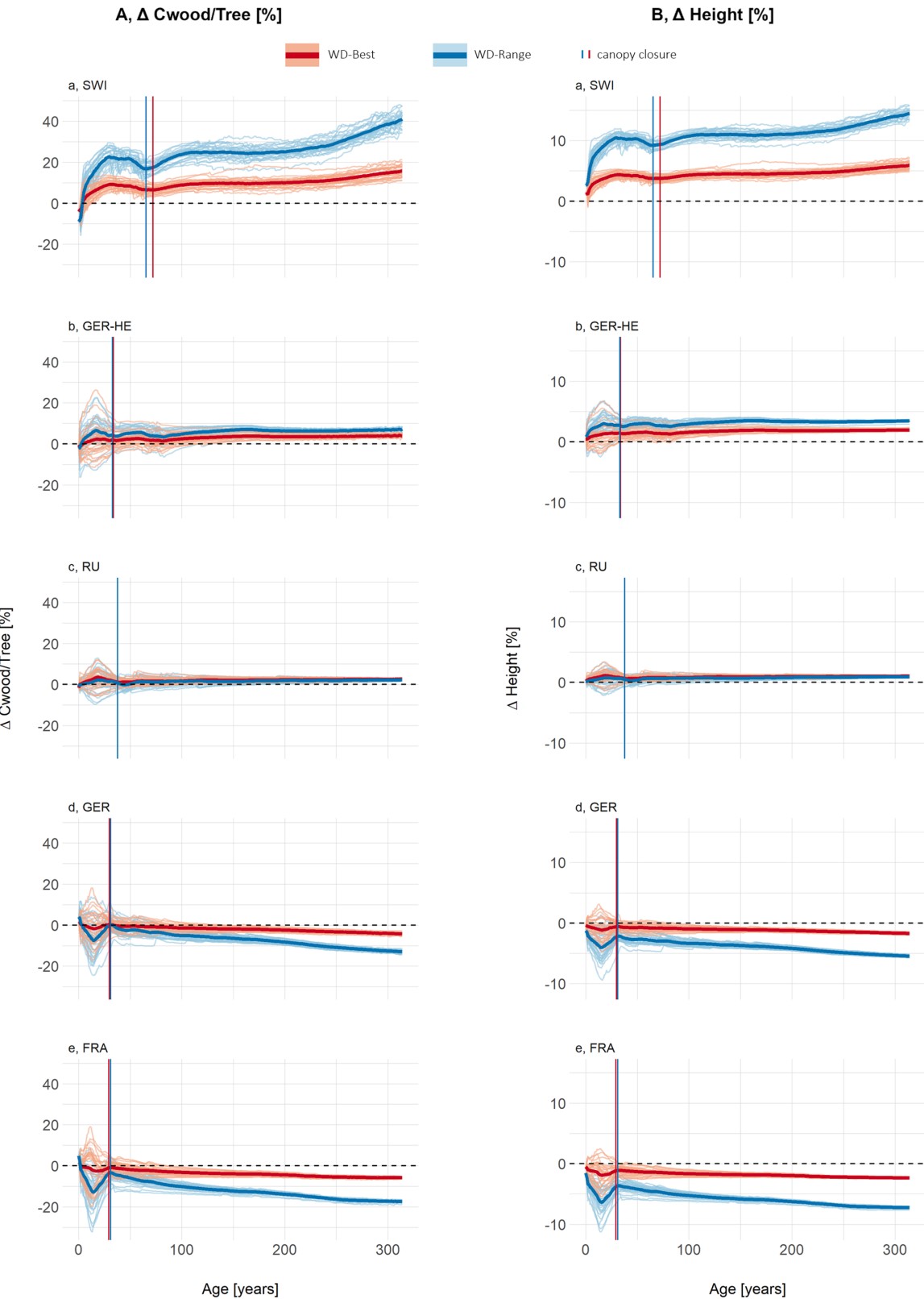

**Figure 3.** (a) Carbon stored in wood per individual tree and (b) mean tree height, both shown as the relative difference between LPJ-GUESS-STD and LPJ-GUESS-WD, using the best-fit (-Best, red) and full-range (-Range, blue) wood density response curves. Lines below the 0-line indicate that LPJ-GUESS-WD results in lower carbon per individual or lower height respectively compared to -STD, while lines above the 0 line indicate higher carbon storage or greater height respectively in -WD simulations. Each lighter line represents a cohort influenced by different climate ensemble members, meaning that climate conditions at establishment vary between simulations. The darker, thick lines represent the mean of all 30 different climate ensemble members. Vertical lines show the mean canopy closure age across all ensemble members. Figure Results- SEQ Figure_Results- \∗ ARABIC 1: (a) The carbon stored in wood per individual tree and (b) mean tree height, both shown as the relative difference between LPJ-GUESS-STD and LPJ-GUESS-WD, using the best-fit (-Best, red) and full-range (-Range, blue) wood density response curves. Lines below the 0-line indicate that LPJ-GUESS-WD results in lower carbon per individual compared to -STD, while lines above the 0 line indicate higher carbon storage in -WD simulations. Each lighter line represents a cohort influenced by different climate ensemble members, meaning that climate conditions at establishment vary between simulations. The darker, thick lines represent the mean of all 30 different climate ensemble members. Vertical lines show the mean canopy closure age across all ensemble members.

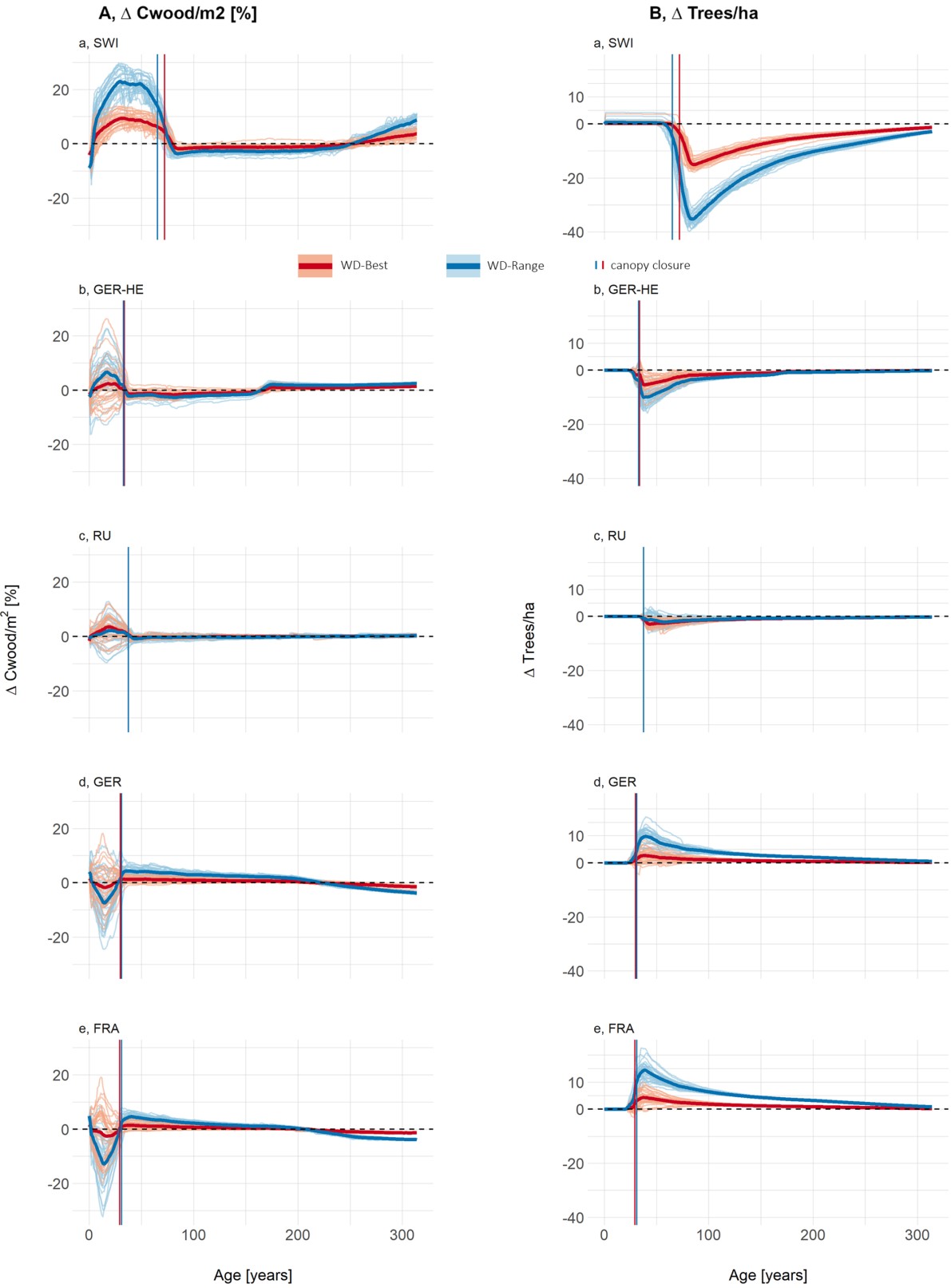

**Figure 4.** (a) The carbon stored in wood per m$^2$, shown as the relative difference and (b) The number of individuals per hectare (ha), shown as absolute difference, both as differences between LPJ-GUESS-STD and LPJ-GUESS-WD, using the best-fit (-Best, red) and full-range (-Range, blue) wood density response curves. Lines below the 0-line indicate that LPJ-GUESS-WD results in lower carbon per m$^2$ to -STD, while lines above the 0 line indicate higher carbon storage in -WD simulations. Each lighter line represents a cohort influenced by different climate ensemble members, meaning that climate conditions at establishment vary between simulations. The darker, thick lines represent the mean of all 30 different climate ensemble.

a new equilibrium across all sites: SWI remains below the LPJ-GUESS standard wood density value, GER-HE reaches it, and RU, GER and FRA exceed it.

Regardless of the age at which climate change was activated, sites displayed a reduction in Cwood/tree (Figure 5a), driven by a decrease in tree height (Figure 5b). One interesting exception is the closed-canopy climate change run at RU, where carbon content increased despite a decrease in height. This suggests that there may be an age and temperature 'window' where WD in the stem, rather than tree height, becomes the dominant driver of an individual's carbon content.

Canopy closure occurred around the same time as for non-climate change simulations, but all climate change simulations were left with more trees. One notable exception regarding the timing of canopy closure is SWI. At SWI, young forests exposed to increased temperatures reached peak ΔCwood/tree much earlier than under baseline conditions (Figure 5a). This suggests that at currently temperature-limited sites, a climate-induced increase in wood density might not immediately reduce forest productivity through allometric constraints. However, this period of high carbon storage relative to LPJ-GUESS-STD was temporary. Over time, the relative difference in Cwood/tree stabilized at levels slightly higher than those observed under cooler conditions.

Across all sites, climate change led to a general decline in ΔCwood/tree, but site-specific trends remained consistent: SWI and GER-HE retained higher ΔCwood/tree compared to -STD, while GER and FRA showed lower values. Compared to LPJ-GUESS-WD without climate change, the relative change in Cwood/tree decreased over the long term, regardless of when the temperature increase occurs. As a result, the absolute difference to LPJ-GUESS-STD in SWI and GER-HE became smaller over time, while in GER and FRA, the difference increased. These trends correspond to the absolute difference between the newly emerging SWD and the LPJ-GUESS-STD wood density parameter.

# 4. Discussion

## 4.1. Wood density influences tree growth rate and thus size

Climate-driven variation in wood density (WD) affected both tree carbon content and stature. In our model experiment, sites where WD emerged as higher than the LPJ-GUESS-STD parameter showed shorter trees, whereas sites with WD lower than LPJ-GUESS-STD show larger trees with less dense wood, supporting H1 that dynamic WD markedly affects tree allocation patterns. While our model is limited to a single species and cohort, the temperature-dependent WD pattern we simulate aligns with broader patterns observed at the community level, where warmer temperatures are associated with higher wood densities (Mo et al., 2024; Yang et al., 2024). Many tree-ring studies have documented an inverse relationship between WD and ring width (Franceschini et al., 2010; Fritts et al., 1991; Larson et al., 2001)—a pattern also observed in the observation data evaluated here (Supplementary Figure S4). This relationship also emerged in our simulations at all sites except SWI (Supplementary Figure S12). Additionally, physiological studies indicate that under unfavourable conditions, such as higher temperatures and increased vapour pressure deficit, trees grow more slowly (Adams et al., 2015) impacting tree height. Furthermore, Augusto et al. (2025) and Gibert et al. (2016) found that across species on site level an increase in WD negatively correlates with the growth rate. The direct effect of WD on tree size is less well-studied (but

see Jucker et al., (2025), though intra-species locally-emergent variability of WD was not assessed). A recent manipulation study (Didion-Gency et al., 2023) found that increased temperature reduced tree height in oak and beech, but the mechanistic role of WD in this process was not investigated. The study attributes this to increased leaf respiration and storage rather than changes in wood anatomy. However, the link between WD and growth increment rates is well-established in ecology, where fast-growing early-successional tree species tend to have a lower WD than slow-growing late-successional species (Chave et al., 2009). The simulated pattern of WD, growth rate, and tree height in this study, modelled annually within a single species, aligns with broader ecological findings. This suggests that the temperature-driven physiological responses we observe may reflect general mechanisms also influencing community-level wood density patterns.

## 4.2. In open-canopy forests, WD has a weak effect on tree density

WD to some extent influenced tree growth rate and thus time to canopy closure. Our simulations predicted that lower WD at cold sites accelerated height growth relative to LPJ-GUESS-STD, causing faster canopy closure and concentrating carbon in fewer but larger trees. In contrast, higher WD slowed height growth, delaying canopy closure and resulting in a shorter forest with more trees. However, we can only partially accept H2, as this impact of dynamic WD on the number of individuals is present, but weak at 3 out of 5 sites. To confirm H2 as a universal and significant effect, the model should be applied to a broader range of species and geographic locations. Nevertheless, a stronger effect on tree number can be confirmed under climate change simulations (H3). There, climate change enforced under open-canopy conditions causes the number of trees to change more strongly relative to LPJ-GUESS-STD. Observed forest growth has been found to be particularly sensitive to the environment directly after regeneration (Itter et al., 2017) which this model replicates. Forest canopy closure levels and not wood density seem to have the strongest effect on forest response to climate change. This is confirmed by several other studies where canopy closure contributes to the stabilization of various forest variables, including microclimate temperature (De Frenne et al., 2019) species composition (Bhatta & Vetaas, 2016).

Alterations of the number of trees have two implications. In terms of carbon dynamics, for example in Switzerland and France, spruce forest tree density ranges between 150 and 300 trees per hectare, depending on the site and forest age (Cuny et al., 2025; Martínez-Sancho et al., 2020). A loss of 20 trees /ha in this context equates to a 6.7%–13.3% reduction in density, which is noticeable and besides forest carbon content and dynamics, could also impact various forest ecological processes.

## 4.3. Current dynamic WD model

The current dynamic WD model is based on a semi-empirical relationship between temperature and LWD. Ring width is emergent from carbon demands of the proposed annual WD, productivity and tree allometry. Studies show that even within a single species, correlations between ring parameters and the environment are complex, varying during the growing season and across biogeographic zones (Björklund et al., 2017; Rathgeber et al., 2016) and over time (Pretzsch et al., 2014). Such spatiotemporal complexity can only be addressed through mechanistic modelling that represents the underlying physiological and developmental processes driving wood formation.

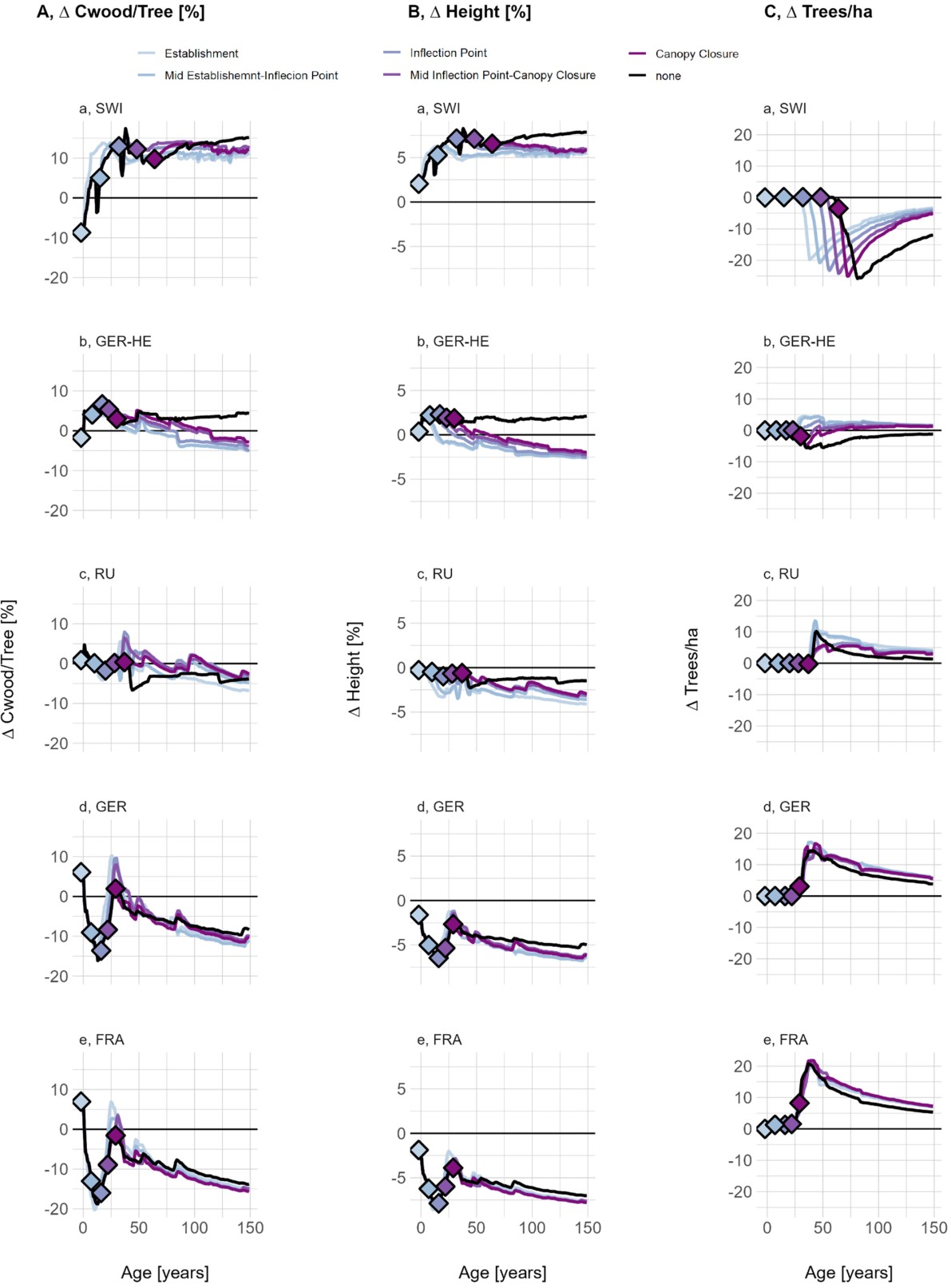

**Figure 5.** (a) Carbon stored in wood per individual tree, (b) mean tree height and C, the number of trees per hectare, all shown as the relative difference between LPJ-GUESS-STD and LPJ-GUESS-WD, using the full-range (-WD-Range) wood density response curve. Different colours indicate simulations varying with respect to the canopy closure stages at which a temperature increase of 2°C occurred. Canopy closure stages were derived as described in the Supplementary materials (Supplementary Section S1.6). Lines below the 0-line indicate that LPJ-GUESS-WD results in lower carbon per individual, height or density of individuals compared to -STD, while lines above the 0 line indicate higher carbon storage, height and density of individuals in -WD simulations.

Our model builds on one of the most robust dendroclimatological relationships: the correlation between LWD and late-summer temperature, extensively employed in paleoclimate reconstructions (Briffa et al., 1998; Briffa et al., 2002; Wilson et al., 2005) and biologically underpinned by the hypothesized breakdown of compensatory hormonal mechanisms during latewood formation (Cuny & Rathgeber, 2016). A key consequence of our model assumptions is transferring this LWD-temperature relationship to TRD via a constant 70:30 (Düthorn et al., 2016) earlywood-to-latewood ratio and fixed EWD. This ratio is based on observations showing that Norway spruce latewood fractions remain consistently smaller (16%–38%) than the earlywood fraction (Sandak et al., 2015). Fixed EWD is a simplification justified by observed stability under moderate climate variation (Cuny et al., 2019), probably due to its hydraulic role (Domec & Gartner, 2002). Considering that we use 30-year repeating climate for H1 and H2, the constant assumptions remain parsimonious for this proof-of-concept study.

We recognize that our current approach of linking temperature effects on LWD to total ring density via a constant earlywood-to-latewood ratio and non-variable EWD likely overestimates TRD responses compared to direct observations. One large-scale study documents declining EWD and increased earlywood-to-latewood ratios in Europe, which besides elevated temperatures is potentially driven by nitrogen deposition and extended growing seasons, (Pretzsch et al., 2018) something our model cannot currently account for. After a period of growth increase, for the above-stated reasons, several regions in Europe now show a decline in growth trend, attributing this to heat, drought and decline in nutrient status (Laudon et al., 2024; Pretzsch et al., 2023). Whether these declines in growth trend are accompanied by changes in WD, and how such changes would feed back on stand dynamics, remains an open question that our modelling framework could help address in the future.

Ultimately, the primary aim of this proof-of-concept study is not to precisely model TRD–temperature relationships, but rather to explore how environmentally responsive WD cascades through carbon allocation, tree growth, competitive dynamics, and forest structure. By modelling this cascade explicitly, we provide a foundation for assessing WD feedback across diverse and changing environments, where direct empirical correlations may not hold.

Future development should target variable earlywood width and density, with water availability as a key intra-annual driver (Bytebier et al., 2022; Camarero et al., 2017) and nitrogen as long-term driver and site-specific modifier. Existing approaches, as proposed by Eckes-Shephard et al. (2021) offer promising directions. Incorporating additional environmental variables and more sophisticated wood formation mechanisms could expand applicability and enhance mechanistic realism.

### 4.4. Usefulness for global vegetation modelling

A model with dynamically emergent WD provides several opportunities to enhance ecological realism and improve carbon cycle representation in vegetation models. First, it allows for a more nuanced simulation of resilience or susceptibility to disturbances by incorporating the legacy of intra-tree variation in WD driven by environmental conditions and life-history. For example, tree size and WD interact to influence resistance to wind damage (Fournier et al., 2013) while trees growing in dry environments tend to develop denser wood, improving drought resilience (Serra-Maluquer et al., 2022). Furthermore, spatially emergent wood density improves the realism of environmental responses and

carbon distribution. In current models, PFTs are typically assigned a single WD value, despite substantial regional variation, such as that observed in the Amazon (Mitchard et al., 2014). Allowing WD to emerge dynamically within PFTs enables a more accurate representation of these differences. Observational studies show that carbon density in PFTs should not be derived from a single parameter or inferred directly from growth increments, supporting a more flexible approach (Pretzsch et al., 2018). While our model focuses on a single species, incorporating dynamic WD may help improve the representation of long-term carbon storage, a relationship supported by global studies linking community-level wood density patterns to tree longevity (Mo et al., 2024). Mean carbon residence time in trees is a key source of uncertainties in forest carbon budgets (Friend et al., 2014; Pugh et al., 2020) and incorporating WD into carbon turnover estimates could reduce these uncertainties. Moreover, considering WD in biomass decomposition models could lead to more realistic simulation of global litter distribution (Brovkin et al., 2012), further refining estimates of forest carbon dynamics.

### 4.5. Source-sink dynamics

The inclusion of dynamic wood density in LPJ-GUESS introduces a new capability to investigate the sink-hypothesis (Körner, 2003). In LPJ-GUESS-WD, wood density is directly influenced by temperature and independent of carbon availability, partially integrating the temperature sensitivity of stem growth as proposed and previously implemented by Leuzinger et al. (2013). Changing the allocation to focus on carbon allocated to the stem, including limiting factors for stem width growth such as temperature and water availability, could further improve carbon storage estimations in trees. In most DGVMs, carbon allocation to the stem is primarily constrained by available carbon. However, some studies suggest that incorporating additional growth limitations independent of carbon availability can improve model realism (Franklin et al., 2012; Leuzinger et al., 2013; Merganičová et al., 2019).

### 4.6. Concluding remarks

To our knowledge, this is the first study to explore the impact of environmentally emergent wood density on tree carbon, allometry, forest structure and carbon content within a DGVM, a key gap in current vegetation modelling.

Our results show that temperature-sensitive wood density triggers cascading effects on multiple forest variables, including tree height, tree number and overall carbon storage. As temperatures rise, higher WD reduces tree-ring width and slows height growth, delaying canopy closure. This, in turn, modifies forest carbon distribution by preventing the rapid concentration of biomass in fewer large trees, leading to lower forest turnover rates and influencing long-term carbon storage patterns.

The magnitude and direction of these effects vary with geographic location and climate, with young forests (prior to canopy closure) being the most affected. Given that approximately 60% of the world's forests are classified as secondary (FAO, 2010; Mackey et al., 2014) integrating dynamic WD into DGVMs enhances the realism of regrowing forest structure, dynamics and resilience.

This study highlights the importance of incorporating environmentally driven WD trait variability into DGVMs. Our conceptual demonstration and implementation explore the potential magnitudes and directions of WD effects, representing a step towards more realistic trait-based modelling of forest carbon dynamics under climate change.

**Open peer review.** To view the open peer review materials for this article, please visit http://doi.org/10.1017/qpb.2026.10038.

## Acknowledgements

The authors are grateful to the anonymous reviewers for their insightful comments, which significantly improved the methodological rigor and clarity of the manuscript. They further want to thank Thomas Pugh and Camille Volle for their contributions in the early stages of this study. They acknowledge the use of artificial intelligence–based tools for limited support in language and presentation, without influencing the scientific content or conclusions of the work.

**Competing interest.** The authors declare no potential competing interest.

**Data availability statement.** All data and code used in this study are available from the corresponding author upon request. DOi: https://doi.org/10.5281/zenodo.18731465.

**Author contributions.** A.-K.V. conceptualized the study and was responsible for drafting the research framework, developing the methodology, processing data, modifying the model, executing model runs, analysing and interpreting model outputs, and representing the results. A.-K.V. also contributed significantly to the manuscript writing. S.O. contributed to the methodology, model development and modification of the model code. S.O. also conducted model runs, participated in result interpretation and provided critical review and comments on the manuscript. H.Z. contributed to the model code and reviewed the manuscript. P.F. provided substantial feedback and expert insights, as well as reviewed the manuscript. A.H.E.-S. co-conceptualized the study and contributed to the research design, methodology, model development, and analysis and interpretation of model outputs. A.H.E.-S. contributed significantly to the manuscript writing.

**Funding statement.** A.-K.V. acknowledges funding from the European Research Council (ERC) Advanced Grant no. 101096708 PERENNIAL. S.O. acknowledges support from the 'ModElling the Regional and Global Earth system' (MERGE) and ClimbForest (grant no. 101059888). H.Z. acknowledges support from the China Scholarship Council (CSC) and a stipend was created to finance the amount not covered by the CSC by the Department of Physical Geography and Ecosystem Science, Lund University. P.F. acknowledges support from the Swiss National Science Foundation project CALEIDOSCOPE (grant no. 212902). A.H.E.-S. acknowledges funding from the VR Etableringsbidrag, grant no: 2024-04678. A.H.E.-S. and S.O. further acknowledges funding from the European Research Council (ERC) under the European Union's Horizon 2020 research and innovation programme (grant agreement no. 758873, TreeMort). This research was supported by the ForestValue programme, the European Commission, Vinnova, the Swedish Energy Agency and Formas for the project FORECO (grant no. 2021-05016). The research presented in this paper is a contribution to the Strategic Research Area 'Biodiversity and Ecosystem Services in a Changing Climate' (BECC) 'ModElling the Regional and Global Earth system' (MERGE) and the Nature-based Future Solutions profile area.

**Supplementary material.** The supplementary material for this article can be found at http://doi.org/10.1017/qpb.2026.10038.

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
