## [Reviewer Report]

In this study the authors explore the effects of considering non-constant wood density (WD) in the dynamic global vegetation model, LPJ-GUESS. Specifically, they implement functions whereby wood density varies with site temperature, both across a species range and also within a site, in parallel to seasonal shifts in temperature.

A key (an important) result was that sites with lower WD tended to develop taller trees, whereas sites with higher WD tended to develop shorter trees – which, all else equal, feels like it has to be the case, the way the maths works in this model for representing relationships between growth, height and WD. Less intuitive (to me) was the finding that sites with higher WD also had more trees (that are smaller) and, then, overall, stored less carbon compared to the control.

The over-arching research question is important, and timely. The methods appear sound (note, I am not a modeller; other reviewers would need to comment on that). Results are clear. The manuscript is clearly written. The quality of figures could be improved somewhat.

In Abstract, lines 16-17, the authors write “Our results highlight the need to incorporate wood density dynamics in DGVMs to improve predictions of forest carbon storage”. In doing so they inadvertently point to the key aspect of the study needing strengthening/justification and perhaps further investigation: without any ground-truthing we don’t actually know that these new predictions will be better than what came before, using the standard representation of WD in LPJ-GUESS. Indeed, it’s also possible – I suppose – that the new model produces worse results in some situations. How would we know, one way or another?

Again, remembering that I‘m not a modeller myself, I wondered whether the authors are sufficiently self-critical. In Introduction, line 72 they write “Incorporating dynamic wood density into DGVMs is therefore essential for improving model predictions and capturing more realistic forest-climate interactions”. “Essential” is surely oversell at this point. The point of the study is to numerically investigate *whether or not* it makes any demonstrable difference to incorporate dynamic WD into the model.

Otherwise, I have little to note. I like the study.

• what is the difference between early wood and latewood other than the part of the season that these tissues are built? Do these tissue types differ in cell sizes, fractions of cell types, functions, etc? Does this distinction between early wood and latewood apply equally to evergreen and deciduous trees? The authors assume the reader is familiar with these concepts but this reader needed more explanation...

---

## [Reviewer Report]

The paper “Environmentally-dependent wood density influences forest structure and dynamics in a demographic vegetation model” explores the effects of including temperature-dependence of a key plant trait (wood density) in the dynamic global vegetation model (DGVM) LPJ-GUESS. It models wood density as dependent on late summer temperature through a logistic curve and shows that this has a number of effects on ecosystem structure and functioning.

The manuscript is well-written, the way that the temperature-dependence is implemented makes sense to me, and the results are interesting, suggesting that trees in warmer areas are more numerous and denser in their wood, but also shorter, leading to a net loss in carbon. Some of these effects come about in the way that carbon allocation is simulated (cf. formulas in the Supplementary that link height and wood density), others are emergent. If true, that would have important ramifications for assessing the global carbon cycle under climate change. The paper would thus be of great interest to readers of Quantitative Plant Biology.

I do have, however, one main concern, which concerns the construction, interpretation and validation of the temperature dependence of wood density, and few minor concerns, which I will detail below, before providing line-by-line comments.

A: Relationship between wood density and temperature (model fitting / validation)

A key assumption of the paper is that late wood density (LWD) is correlated with temperature at the individual tree level, and that this association can be directly transferred to tree ring density (TRWD), because early wood is assumed to have a constant wood density and early wood late wood ratios (EW/LW) are assumed to be constant as well. However, the paper does not currently provide adequate proof for these assumptions – the main reference of EW/LW staying constant is a single study with trees exclusively from Finland and with limited sample size. I see several issues.

1. Tree ring density: The authors have access to some tree ring density data (Figure S3), and the model only seems to rely on TRWD (cf. Supplementary). So even if the available TRWD data cover a smaller geographic extent, the authors should directly model TRWD as a function of temperature. At the very least, this should be provided as a Supplementary, but it makes more sense to report this as the main formula, and provide a supplementary formula for cases where only LWD is available. Figure S2 and S3, as currently provided, do not clearly show that the hypotheses are valid.

2. Simpson’s paradox: This seems common in dendrochronological studies, but there is a danger of mixing/confounding non-dynamic effects across species and individuals with dynamic effects within individuals. For example, in the Supplementary, the authors plot Picea abies and obovata together, but these two species likely differ in their overall wood densities and occur in different regions, so part of the temperature effect could be created by biogeographic patterns and not reflect a dynamic response (e.g., among-species effects differ from within-species effects). Even more importantly, all sites and individuals are mixed together. Here as well, the effect across sites and individuals does not have to correspond to the effect within sites and individuals. Separating the species, sites and individuals may not change the observed relationships, but there is ample potential for Simpson’s paradoxes / ecological fallacies here, and the authors need to show that this is not the case. This can be done by simply plotting correlations separately for the two spruce species, the different sites, and also plotting correlations at the individual level. One can also mean-center species, sites, or individuals and then plot them together. For fitting the wood density-temperature relationship, it is possible to use mixed-effects models to account for between-species, between site, and between-individual variation.

3. Wood density effects within species vs. communities: The previous point goes along with an ambiguity in interpretation. The authors frequently cite studies that have shown community-level shifts in wood density with temperature (i.e., due to changes in species composition), but evaluate wood density shifts within species. I would urge the authors to more clearly separate the two.

4. Validation: The authors mention the study by Pretzsch 2018 in the beginning – a study that shows decreases of wood density in the past centuries, i.e., under increasing temperature –, but do not discuss / validate their models against this study or alternative data. Do temperature correlations with LWD really translate into correlations with stem wood density? Just as an example, when I used a collation of openly available data – e.g., from Martinez-Sancho et al. 2020 (10.1038/s41597-019-0340-y) and Schepaschenko et al. 2017 (10.1038/sdata.2017.70), standardized to basic density –, and plotted measured stem wood densities for spruce against temperature or assessed wood density by country, I did not find any evidence for an increase with temperature. Database collections can be confounded by measurement errors, so I am not saying that this is definitive evidence, but all the evidence in this collection points to stability or decreases in wood density with temperature in spruce trees rather than the opposite, and it would be good to harmonize this evidence with the authors' assumptions/findings.

5. Wood density absolute values: this is more of a side note, but I generally found wood densities in this study to be at the high end of what we would expect for Picea abies (typical densities of 0.35 g cm-3, so ca. 0.17 gC cm-3, but the study reports average wood densities of 0.22 gC cm-3 and up to 0.25 gC cm-3 under temperature increases, which would be ~0.5 g cm-3).

In general, I would have expected a more overarching Hypothesis (H0, or H1) that assesses the validity of the model.

B: Figures

I generally thought the Figures were nice and intuitive, in particular with the indication of canopy closure, but the authors could improve them somewhat to make them more easily readable (e.g., remove the coordinates of the sites, add column headers, maybe remove the legends for the “Ensemble Members”, which are quite easy to understand). Also, if the sites could be narrowed down to 3 sites for the main text (e.g., SWI, GER, GER-HE), it might be easier to read the figures, but that’s just a suggestion.

C: “Individual trees”

The manuscript frequently describes processes as if they happen at the level of individual trees in the DGVM. Unless I am mistaken, this is not the case. Individuals are bound up in cohorts and cannot differ from other individuals in the same cohort. So everything happens at the level of the cohort, and what changes is an “average individual”. I don’t think that’s problematic, but I would strongly suggest to change the terminology, because individual tree level changes suggest that individual trees can develop separately from each other, which is not the case here. But I may have misunderstood how trees are simulated in LPJ-GUESS, in that case ignore this point.

Line-by-line:

5: The abstract is nice, but I would suggest making it a bit more quantitative: how much (% and/or in absolute values) does it change predictions for forest carbon storage, forest carbon dynamics, tree numbers, tree height, etc.

8: environment-dependent

14-15: the last part of the sentence is probably not needed

15-16: I don’t understand this. What does it mean that the stage before canopy closure is relevant to most of the world’s forests? Do you mean that most forests are in this stage? What about dense tropical forests?

20: The introduction is generally well written, and easy to understand!

26-28: Maybe rephrase – this makes it sound like forest carbon storage is responsible for humanity’s fossil fuel emissions

32: Is this really at the individual tree level? Tree growth-strategy sounds more like species level / evolutionary strategies

38-40: “affecting” is maybe too uniformly causal – it is certainly linked to all these properties, but in some cases it may be summarizing other traits that are more directly affecting them (e.g., fibre density, etc.)

46: I am not sure these are the best references for this, especially for variation among species; consider, for example, Phillips 2019, Surveys in Geophysics (https://doi.org/10.1007/s10712-019-09540-0); Rosas et al. 2019, New Phytologist (https://doi.org/10.1111/nph.15684); Anderegg et al. 2021, New Phytologist (10.1111/nph.16795 )

79-81: I think it’s fine to model wood density as temperature dependent for a first pass, but this needs to be justified more. Why not water availability, why not wind dependence, why not disturbance dependence?

83-85: Is this the best justification? We know that wood density is comparatively strongly controlled genetically, i.e., intraspecific variation or biogeographic shifts are probably not as important as community turnover due to environmental shifts. A better example may be shifts within a PFT, e.g. a shift within conifers from spruce in Germany to larch in Siberia, and the associated wood density shifts.

91: I was missing a hypothesis (H0) that your model of temperature dependent wood density is accurately modelling the qualitative behaviour of observed wood densities. To me that’s a precondition for including it in the model

94: This hypothesis makes sense, but I struggle with the “individual level” – this makes it sound as if the individuals in the same plot vary, but as far as I understand up to that point, that’s not what is simulated, right?

111-112: Cf. my concerns above – is this such a good study system? Should you not rather look at species turnover between communities?

113-114: Sometimes, the manuscript describes processes as if they happen at the level of “individual trees”, but if I understand this correctly, it is rather cohorts of trees, where trees within cohorts cannot differ from each other. I would suggest reformulating the respective passages, maybe using something like “cohort” or “average tree” or “typical tree”, because “individual trees” suggests that trees can differ from each other.

121-128: This is nice!

126: Nice, so LPJ-GUESS-STD explicitly models rings, including EW and LW?

138: Is this really a feedback between an individual tree’s carbon content or between a cohort’s carbon content?

165-198: Cf. my wider points above. I think these hypotheses would need more testing / proof than is currently provided. A lot in this study hinges on the assumption of constant EW/LW ratio, which is based on a small study from Finland, and non-dynamic EW. But is this supported by the literature? Björklund et al. 2017, for example, found a negative association between ring width and earlywood density and a positive association between ring width and latewood density, which suggests to me that EW cannot simply be taken as constant (i.e., it may negatively correlate with LW). The authors do have tree ring density measurements (Figure S2), so they should plot correlations between temperature and tree ring density (and not just latewood density), and they should rule out Simpson’s paradoxes by also checking the relationships for the two different Picea species separately, for the sites separately, and for the individual trees separately. Figure S3 is not sufficient for this purpose (198).

196-198: I am not convinced by this yet.

200-207: Cf. above. I would like to see this graph with tree ring density.

212: Cf. above. This value seems to be at the high end for Picea abies (I would have expected something around 0.17-0.18 kgC / m3), same in Table S3 later.

235: That’s quite heavy Picea abies (0.5 g cm-3)

288 and 318: Nice figures! So if I summarize – sites with lower wood density (SWI / GER-HE) than the average have taller trees, more carbon, and fewer individuals. That trees in hotter areas are shorter and have higher WD makes sense, but I am wondering about the higher tree densities. Is this something we observe in the field with Picea abies? Would you have any validation data to test this against?

315: There is something missing

367-369: This goes back to my earlier point about Picea abies. If the study was reframed as being about community shifts instead of within-species shifts, I would agree, but otherwise these studies show community-level wood density shifts with temperature, not within-species shifts.

---

## [Editor Report]

Your manuscript has been fully evaluated by two independent peer reviewers with complementary backgrounds.

Both reviewers underlined the importance of your results and appreciated that your manuscript is well-written and nicely illustrated. However, they stressed the need for better explaining essential concepts (e.g. earlywood / latewood) and better justifying key hypotheses (e.g. the correlation between late wood density and temperature at the individual tree level). Another point raised is the validation of your predictions. Could you strengthen your results with such validation?

I look forward to receiving your revised manuscript in which you carefully address all points raised by the reviewers.

---

## [Reviewer Report]

As I noted in the initial review, this is an interesting piece of work. It’s still not clear to me how best to demonstrate that the new approach to modelling WD is better than what came before (nor if the authors do this) but, in principle, it should be.

---

## [Reviewer Report]

This is a review of the revised version of “Environmentally-dependent wood density influences forest structure and dynamics in a demographic vegetation model”. Since I have commented on the writing style, relevance and main points of the manuscript before (Reviewer 2), I will primarily address the revision. I still think that the manuscript is well-written, tackles an important question and generally uses an elegant approach, and I thank the authors for their detailed point-by-point responses. The authors have addressed the reviewer comments in great detail and have generally made appropriate adjustments to the text, including the Supplementary. I particularly like the figure that plots WD/TRD values against temperature in the response document, and the detailed analyses of how temperature - LWD relationships hold across hierarchical levels. That is great work, and it’s great to see that the relationship holds at various levels of aggregation, pointing to a mechanistic response. Thanks for putting so much effort into this!

However, I still have three concerns: 1. The proof-of-concept nature of the study should be made more explicit. This can be done by including explicit statements to this effect, caveating results, and potentially using past tense for the results, which would make them appear less general. 2. Based on the new data the authors show, the modelled WD-temperature response function seems to strongly overestimate the empirically observed WD response to temperature, and this should be made more explicit. I would suggest including the empirical data and empirical response function in Figure 2 (provide the same figure as in the response to reviewers) and discussing this more explicitly in the text. 3. I believe there is a conceptual error in how the study converts wood density values into carbon estimates, essentially treating airdry wood densities as basic densities. I only discovered this when reading the response to reviewers. If true, this would require a revision and, unfortunately, might necessitate a rerun of models. I will provide details below.

1. Proof-of-concept study

In the authors’ responses to reviewers, they argue that the manuscript is intended as a proof-of-concept/sensitivity study to showcase how important the modelling of wood density variation could be for modelling carbon and demography. This is a valid aim, and the study will make a great case for it. Some of the choices in the manuscript – picking data from a limited geographic scope, modelling an extreme wood density response which deviates from the one directly observed in the data, not explicitly validating the demographic predictions – also make sense in this context.

However, the “proof-of-concept” nature of the study is not sufficiently clear in the current manuscript. Many parts of the manuscript have been written as if they were definitive results. E.g., in the abstract (l.13-15), the authors state that “sites with higher wood density have more but smaller trees and store less carbon compared to the standard model. The strongest effects occur before canopy closure, a stage currently characterizing most global forests, with tree carbon deviating by up to 33%.” To me, this sounds more definitive than a proof-of-concept study and incentivizes readers to cite this study as evidence of observed wood density effects on carbon storage. I believe that this lack of clarity about the proof-of-concept nature explains why both Reviewer 1 and I previously raised validity concerns.

Given my still persisting validity concerns (cf. below), the authors should highlight the proof-of-concept nature of the manuscript more clearly throughout the abstract and text of the manuscript. This way, readers would better understand the preliminary nature of the investigation. This probably only requires few adjustments, and I have made several suggestions to this effect in the line-by-line comments below.

2. Validity

Even if the study is intended as proof-of-concept, validity concerns play a role, and I still don’t follow the authors’ justifications for modelling WD/TRD through temperature effects on LWD and a constant EW-LW ratio instead of directly modelling TRD in response to temperature. The authors’ responses/data show that their approach is likely strongly overestimating temperature effects on wood density. The most likely explanation is that the constant EW-LW ratio assumption is violated and that the EW-LW ratio is itself responding to temperature, thus compensating for increases in LWD with temperature. In the following I will address the authors’ main points:

1. Weakness of temperature signal on TRD: In their responses to reviewers, the authors provide a beautiful key figure which actually plots observed TRD against temperatures. I find this plot immensely valuable for the vegetation modelling community and would advocate to make this Figure 2/Figure methods-2, including all data points and the direct TRD temperature response (green line), even if it is not used in the model. If this figure is shown, then any reader could better understand the authors’ approach and how it may deviate from empirical data. Further, I disagree strongly with the authors statement in response to reviewers: “We see that the temperature signal in the TRD is too weak to use for creating a valuable temperature-density response function.” The entire point of the article seems to model the response of TRD to temperature. The authors have actual data to do this (amazing!) and find relatively small effects (makes sense, as nature often shows small effects, cf. also the recent preprint: https://doi.org/10.1101/2025.08.25.671896). However, they then seem to pick a less accurate response function, presumably because it shows stronger effects? Even in a proof-of-concept study, the main concern should be a response within realistic ranges, not a strong, but unrealistic response.

2. Parsimony: the authors argue that they have chosen a parsimonious approach. I have generally mixed feelings about parsimony, but I can see the point of parsimony when there is no empirical data. However, in this case, there is empirical data, and modelling TRD directly as function of temperature seems more parsimonious to me than modelling LWD as function of temperature and then translating this into TRD by making an assumption about the EW-LW ratio that would require much more testing.

3. Limited geographic scope: One thing I did not appreciate before is that the ITRDB data may be geographically biased to high-latitude and high-altitude sites and thus show particularly strong temperature effects. As part of a proof-of-concept study, it can make sense to pick the most extreme cases, but it should still be made clearer in the manuscript that this may not be representative of the majority of woody communities.

In summary, I think a direct modelling of TRD would be more appropriate, but I will not insist on this approach, and I can see the long-term benefits of separating the modelling of LWD and EW-LW ratios. In this case, however, the authors should make much clearer that they are picking a stronger response function than indicated by the empirical data. The best and most useful way to do this is to include the empirical WD/TRD data and the empirical response function (green line) in Figure 2 in the main text, and explain carefully why a more extreme, if more unrealistic, response function was chosen, and then also discuss this in the Discussion section.

3. Airdry vs. basic density

Unfortunately, I missed this point in my first review. When translating wood density values into carbon, it is important to differentiate between airdry densities (airdry mass / airdry volume) and basic density (ovendry mass / fresh volume). Only basic density provides estimates of biomass per living volume, while airdry densities still include water (typical moisture content of 12-15%). To translate airdry densities into carbon content, they need to be first converted to basic densities by multiplication with a rough factor of 0.83 (e.g., here: https://doi.org/10.1002/ajb2.1175). Then carbon content can be calculated with a factor of around 0.5, as the authors mention (l.190).

On rereading the manuscript and the authors’ responses, I found that the authors do not differentiate clearly enough between different wood density types. In their response to reviewers, they cite wood density values from x-rays calibrated to yield airdry densities (Sandak and Pretzsch studies) in the same context as measurements of basic density. This prompted me to look up wood density measurements in the ITRDB, and as far as I understand, the measurements provided there are actually calibrated to correspond to airdry estimates (is this true?). If this was the case, the authors’ conversion of wood density to carbon content would be incorrect. Instead of using a conversion factor of 0.5, they should use a conversion factor of 0.83 * 0.5 = 0.415. I would urge the authors to verify the type of wood density they obtained from ITRDB and potentially adjust their conversion factors.

Line-by-line comments:

l.8-10: It would be good to make the proof-of-concept nature of the study more explicit early on. I suggest: “In a proof-of-concept study, we explore how temperature-dependent dynamic wood density could impact tree- and forest-level carbon storage by integrating it into the DGVM LPJ-GUESS.”

l.13-15: These two sentences should be modified to better reflect the proof-of-concept nature of the study. I suggest shifting to the past tense (less general) and highlighting the model/predictive nature as opposed to observed facts. My suggestion: “Based on our model formulation, sites with higher wood density had more but smaller trees and stored less carbon compared to the standard model. The strongest effect was predicted before canopy closure, a stage currently characterizing most global forests, with tree carbon deviating by up to 33%.”

L.27: I missed this previously. How do you arrive at 50%? To my knowledge, the data support around 30% (and only when not taking into account deforestation)?

l.83-85: I would strongly suggest being clearer here and say: “In this module, wood density emerges dynamically based on a non-linear response function between temperature and late wood density, …”

l.85: Not quite clear what “Previous” means here. Apologies if I overlooked this before.

l.101: Could you potentially remove “at the individual level” here or change it to “at the level of an average individual per cohort”? I appreciate that “individual” is now defined in 121-123, but here it has not yet been defined, and this makes it clearer to the reader.

l.121-123: this is clearer now! Thanks for including this

l.179-183: Cf. my wider points above. To me, this still does not make sense. Since there clearly is data on tree ring density and the robustness of the constant LW-EW assumption is doubtful, why not model tree ring density directly as a function of temperature? This is by far the most parsimonious decision. My suggestion would be to also show the empirical WD/TRD data in Figure 2 and the empirical response function, and clearly state here that the chosen modular approach actually overestimates empirically observed WD/TRD responses to temperature. This can probably be justified due to the proof-of-concept nature of the study and future developments on the EW-LW ratio, but readers need to understand this.

189-190: This conversion makes sense for biomass, but the densities measured in the ITRDB likely also include 12-15% water content (but correct me if I’m wrong on this!), so the factor of 0.5 is an overestimate of carbon content by ca. 20%. The appropriate conversion would be something like 0.83 * 0.5 (or maybe 0.827 * 0.485), or 0.41-0.42. Also, I would suggest citing the more comprehensive Martin et al. 2018, Nature Geoscience (https://doi.org/10.1038/s41561-018-0246-x).

210-214: The empirical data for WD/TRD vs. temperature do not support a constant 70:30 fraction, as the empirical function (green line) diverts strongly from the modelled functions. There seems to be a compensatory response in LW-EW ratio with temperature.

2016-2020: As suggested earlier, please add the raw data and the empirical response function. This makes it very clear what is assumed in the model, and what is empirically supported.

245: Potentially you could use the past tense for the results. That makes the results sound more like a proof-of-concept and less general.

378: Here, it would be great to start out with a short reminder that this is a proof-of-concept. E.g., “In our proof-of-concept study, sites where WD/TRD was higher than LPJ-GUESS-STD parameters showed shorter trees, …”

381-384: This is nice.

386: suggesting: “also observed in the data evaluated here”

425-426: This is potentially misleading. It is based on an empirical relationship between temperature and LWD, and then makes a strong assumption about EW-LW ratio. This needs to be expanded. The article also needs to discuss why WD/TRD was not directly modelled as function of temperature. Again, maybe mention proof-of-concept nature of the study.

426: “identified more passively” is a bit difficult to understand, maybe “predicted indirectly?”

439-444: I strongly disagree that the constant ratio assumption remains parsimonious. The data sources cited provide rough averages, but the data in this study suggest that the current model does not predict WD/TRD well. Otherwise, how can we explain that the empirical response function (green line in Figure 2 provided to reviewers) does not align with the modelled response function?

445-454: This is a good passage, but would need 1-2 literature sources (especially in the first paragraph) and should be merged with the previous one to clarify the challenge of modelling EW-LW ratio, and also highlight more that the current proof-of-concept likely overestimates WD/TRD response to temperature.

---

## [Editor Report]

Your revised manuscript has been reviewed by the two reviewers who already commented on the original version. Both of them deeply appreciated that you have been putting great effort on reanalyzing data and improving th manuscript. However, reviewer #2, while being very positive on your work, is still asking for revisions. Most of them are essentially about rephrasing and discussion, but others may involve reruning your model, if you deem it necessary. There is a caveat about the definition of wood density (airdry vs basic). Please make sure your way of estimating biomass from density is correct.

I look forward to receiving your new revised manuscript, and I sincerely thank you for all your efforts for producing what will be an excellent published article.

---

## [Reviewer Report]

I thank the authors for their careful revisions and answers. I appreciate a lot the now strong emphasis on the exploratory/proof-of-concept nature, the more detailed discussion of deviations from the best empirical model, and the recalculations from airdry to basic density. This has made the manuscript much stronger, and I think it would be of great interest to readers of Quantitative Plant Biology.

I also understand the authors’ position that they want to explore a mechanistic model and I agree with their overall strategy. It is absolutely valid to pick a more mechanistic model over a (likely overfitted) empirical model. To play devil’s advocate, however, the difference between mechanistic and empirical approaches is generally not so clear cut. In particular, the argument for mechanistic modelling is weakened by the fact that the author’s approach also involves several decisions that seem purely empirical/data-driven. For example, the authors define the WD response with regard to July-August-September temperatures, because that was “the period of highest correlation” (l.205), or define an alternative TRD-Range model, because the original model did not capture the full range of the data.

With this in mind, I would only have one last, minor recommendation: I would ask the authors to also show the raw data points in Figure Methods-2 (just as in Figure S6). I can see the point of not plotting the empirical fit in Figure Methods-2 - although I would find it more interesting to include it -, but I don’t see any good justification for omitting the raw data. It is good scientific practice to show models together with the data they were fitted to, and this would make the descriptions in 192-219 much more understandable to the readers of the article.

Detailed comments:

15: I appreciate a lot that the authors now highlight the exploratory nature of the study. In this sentence they could even just say “Our study” instead of “This exploratory study”. It is now clear already from the remainder of the abstract that this is exploratory.

79-80: maybe full stop and new sentence?

94: I thank the authors again for their revisions to make clear the proof-of concept nature. I think it is now already clear from context, so they could also write: “This study aims to explore the potential implications…”

188: Thanks for this correction! Great work!

417: but also see this recent study where no direct impact of wood density on height was found (once other factors were accounted for): Jucker, T., Fischer, F.J., Chave, J. et al. The global spectrum of tree crown architecture. Nat Commun 16, 4876 (2025). https://doi.org/10.1038/s41467-025-60262-x

460-470: nice!

478-483: Ideally, also guided by data availability?

---

## [Editor Report]

The reviewer is very positive on your revision. And I personally thank you for the great care with which your carried out the successive revisions. I am extremely pleased to endorse now your article for publication in QPB. This is exactly the kind of excellent quantitative work we want to publish in our journal!

If you deem it useful, you can slightly revise your manuscript before publication to take into account the last minor comments of the reviewer. But this is up to you.